# Porcine placenta hydrolysate as an alternate functional food ingredient: *In vitro* antioxidant and antibacterial assessments

**Phanthipha Laosam[1], Worawan Panpipat[1]\*, Gorawit Yusakul[2], Ling-Zhi Cheong[3], Manat Chaijan[1]**

**1** Department of Food Science and Innovation, Food Technology and Innovation Research Centre of Excellence, School of Agricultural Technology and Food Industry, Walailak University, Thasala, Nakhon Si Thammarat, Thailand, **2** School of Pharmacy, Walailak University, Thasala, Nakhon Si Thammarat, Thailand, **3** Zhejiang-Malaysia Joint Research Laboratory for Agricultural Product Processing and Nutrition, College of Food and Pharmaceutical Science, Ningbo University, Ningbo, China

\* pworawan@wu.ac.th

**Data Availability Statement:** All relevant data are with the manuscript and its Supporting information files.

## Abstract

The production of bioactive peptides from animal-based raw materials highly depends on enzymatic hydrolysis. Porcine placenta is an underutilized biomass in Thailand's pig farms, yet it is still a source of proteins and beneficial compounds. Porcine placenta could be used as a protein substrate for the production of enzymatic hydrolysate, which could be employed as a functional food ingredient in the future. The goal of this study was to enzymatically produce porcine placenta hydrolysates (PPH) using three commercial enzymes (Alcalase, Flavouzyme, and papain) and evaluate their *in vitro* antioxidant and antibacterial activity. The degree of hydrolysis (DH) increased as the enzyme load and hydrolysis time increased, but the DH was governed by the enzyme class. The maximum DH was found after using 10% enzyme for 20 min of hydrolysis (36.60%, 31.40%, and 29.81% for Alcalase, Flavouzyme, and papain). Depending on the enzyme type and DH, peptides of various sizes (0.40–323.56 kDa) were detected in all PPH. PPH created with Alcalase had an excellent reducing capacity and metal chelating ability ($p < 0.05$), whereas PPH made with Flavourzyme and Papain had higher DPPH$^\bullet$ and ABTS$^{\bullet+}$ inhibitory activities ($p < 0.05$). Papain-derived PPH also had a strong antibacterial effect against *Staphylococcus aureus* and *Escherichia coli*, with clear zone values of 17.20 mm and 14.00 mm, respectively ($p < 0.05$). When PPH was transported via a gastrointestinal tract model system, its antioxidative characteristics were altered. PPH's properties and bioactivities were thus influenced by the enzyme type, enzyme concentration, and hydrolysis time used. Therefore, PPH produced from porcine placenta can be categorized as an antioxidant and antibacterial alternative.

## Introduction

During the years 2019–2020, more than 300 million tons of primary livestock were produced and processed into meat products around the world [1]. Pig production is increasing

**Funding:** This work was financially supported by the Thailand Science Research and Innovation (TSRI) under the Research and Researchers for Industries-RRI program [Grant Number PHD62I0014] and the New Strategic Research project (P2P), Walailak University, Thailand. The funders had no role in study design, data collection and analysis, decision to publish, or preparation of the manuscript.

**Competing interests:** The authors have declared that no competing interests exist.

considerably in many countries, with worldwide pig meat output anticipated at 109.2 million tons in 2020 [1]. In Thailand, small-scale pig production systems have recently been replaced by more intensive large-scale economically focused pig farms [2]. Wastes and by-products are being reassembled in the farrow-to-finish production pig farming system, which consists of breeding pigs, producing piglets, and fattening pigs, as a result of the expansion of large-scale pig farms [3]. Placenta is one of the valuable biomass generated from the pig production [4–6]. However, there has been no published research on the use of pig placenta in Thailand. It is of great interest to figure out how to use the right technologies to add value to this placenta.

The placenta is a transitory organ that carries nutrients, respiratory gases, and wastes between the maternal and fetal systems during pregnancy. Bioactive molecules (hormones, growth factors, enzymes, bioactive peptides, minerals, and vitamins) are stored in the placenta [4]. Collagen is the most abundant protein in the extracellular matrix of the placenta, with lately high market values [4–6]. Despite their high protein content, placentas are frequently discarded or utilized as fertilizer in Thailand. Because porcine placenta is typically discarded, greater efforts are needed whether this by-product may be used. Several research have indicated the possible use of animal and human placenta extracts with high biological and antioxidant activity. Tang et al. [6] suggested that the goat placenta extract was primarily composed of peptides having a molecular mass of 3–10 kDa. Peptides with high quantities of glycine, glutamic acid, aspartic acid, and tyrosine were discovered to have antioxidant activity [7]. Laennec (Japan Bio Products, Tokyo, Japan) is a human placenta extract that is currently commercially available as an injectable for the treatment of climacteric symptoms and chronic varicose ulcers [8–10]. Moreover, enzymatic hydrolysis-prepared placental peptides had excellent bioactivity [7], demonstrating that enzymatic hydrolysis is effective in liberating bioactive peptides from the parent placenta. Typically, the structure of peptides produced, which is influenced by the type of proteolytic enzyme, the degree of hydrolysis (DH), and processing conditions, determines the techno-biofunctionality and biostability of protein hydrolysate [11]. Because the type of protease utilized affects the type of peptides produced, there is a lot of interest in figuring out how the structure-bioactivity interplay of peptides is altered by enzyme type. The synthesized parameters are also important for understanding and designing prospective tailor-made peptides for certain applications. Enzymematic hydrolysis is one of the most commonly used methods for creating bioactive peptides from protein-based sources. When compared to microbial fermentation, it has the advantages of being easier to scale up and having a shorter reaction time. Previous studies have shown that different endoproteases and protein substrates can be used to make protein hydrolysates that can be used in certain formulations [12]. Because protease's ability to breakdown polypeptide substrate varies greatly, choosing the right enzyme to acquire the desired nutritional characteristics and bioactivities is challenging [13]. The effects of trypsin, chymotrypsin, and pepsin on the hydrolytic capacity of thermally treated porcine placenta were explored by Jung et al. [14], and trypsin was found to be the most effective in hydrolyzing porcine placenta. Alcalase, Flavouzyme, and papain were among the commercial proteases studied extensively for creating protein hydrolysates from a variety of protein substrates. *Bacillus licheniformi* and *Aspergillus oryzae* provide Alcalase and Flavorzyme, respectively, while papain is derived from papaya latex. Specifically, papain is an endoprotease that breaks the peptide bond between the serine and cysteine positions in polypeptide chains. Alcalase is a serine endopeptidase that can produce a combination of short peptides [15]. Flavourzyme is a mixture of endoprotease and exoprotease with the capacity to break polypeptides between and at their ends with broad specificity [16]. As a result, distinct peptide sequences with variable chain lengths can be produced by hydrolyzing different protein substrates with these three proteases. The usefulness and bioactivity of tailor-made peptides will be determined by their molecular structure [17]. Clemente [12] suggests that one of

the most important factors determining the nutritional properties and bioactivities of the resulting protein hydrolysate is the source of protein substrate. As a result, nutritional properties, bioavailability, bioaccessibility, biostability, and bioactivity of resulting peptides may be affected by enzyme type and protein substrate supply. To meet the specific benefit criteria, it is still challenging to define the optimal hydrolysis condition in diverse enzymes and protein substrates. There have only been a few reports on the comparative activity and efficiency of these three enzymes on porcine placenta, which has been underutilized in Thai pig farms. As a consequence, the goal of this study was to compare the impact of proteases (Alcalase, Flavourzyme, and papain) on the hydrolysis efficiency of porcine placenta hydrolysate (PPH). The various antioxidative properties, as well as their association with hydrolysis parameters and antibacterial activity, were studied. PPH's antioxidative behavior in a GI tract model system was also investigated in order to see if it may be used as an alternative functional food ingredient.

## Material and methods

### Chemicals

All enzymes (Alcalase from *Bacillus licheniformis*, Flavourzyme from *Aspergillus oryzae* protease, papain from papaya latex) and chemicals (e.g., trichloroacetic acid (TCA), 2,4,6-trinitrobenzene sulfonic acid (TNBS), 2,2-diphenyl-1-picrylhydrazyl (DPPH), 3-(2-pyridyl)-5-6-diphenyl-1,2,4-triazine-4′,4″-disulphonic acid sodium salt (ferrozine), 2,2-azino-bis(3-thyl-benzothiazoline-6-sulphonicacid) diammonium salt (ABTS), 2,4,6-tripyridyl-s-triazine (TPTZ)) were purchased from Sigma-Aldrich (St. Louis, MO, USA).

### Placenta handling and basic composition analyses

One hundred kg of placentas from crossbred sows (Landrace × Large white × Duroc (LLD)), with the same genetic background and similar feeding/housing conditions, was obtained from Shaw Kaset Rungrueng Co. Ltd., Nakhon Si Thammarat, Thailand. The collected placentas were washed twice with cold water (4˚C) to remove blood and mucus. Washed placentas were cut into small pieces, ground with a Talsa Bowl Cutter K15e (The Food Machinery Co., Ltd., Kent, UK) to generate a homogeneous/composite sample, and lyophilized using a freeze-dryer (FTS systems Inc., Stone Ridge, NY, USA). The proximate composition [18], TCA-soluble peptide [19], mineral profile (zinc, phosphorus, iron, manganese, magnesium, calcium, copper, sodium, and potassium) [20], and amino acid contents [20] of the freeze-dried samples were all determined. After vacuum packing at -60˚C, the remaining freshly ground porcine placentas were kept for enzymatic hydrolysis.

### Preparation of porcine placenta hydrolysate (PPH)

Ground porcine placentas were hydrolyzed with three commercial proteases at different enzyme loading of 0, 2.5, 5.0, 7.5, and 10.0% (w/w of placenta protein) at 50˚C. For each enzyme, the following hydrolysis mediums were used, each with its own optimum pH: Alcalase: 0.05 M glycine-NaOH buffer (pH 9.5), Flavourzyme: 0.05 M phosphate buffer (pH 7) and papain: 0.05 M phosphate buffer (pH 6.5). Ground porcine placentas were homogenized with 10 mL of individual cold buffer using an IKA® homogenizer (Staufen, Germany) to obtain 10% protein concentration. The mixture was incubated in a Memmert controlled water bath (50˚C) (Schwabach, Germany) with continuous stirring at 200 rpm. The sample was taken at 0, 5, 10, 15, and 20 min to evaluate the α-amino acid content, DH, and antioxidative properties after enzyme inactivation (95˚C/ 20 min). The solution was centrifuged at 4,500 ×g (4˚C) for

20 min using an RC-5B plus centrifuge (Sorvall, Norwalk, CT, USA). Then, the supernatant was collected and used for further investigation.

## Measurement of α-amino acid (AA) content and DH

The AA content and DH of PPH were determined using the methods of Nissen et al. [21] and Sripokar et al. [22]. PPH (250 μL) was mixed with 0.2 M phosphate buffer, pH 8.2 (2 mL) and 0.01% TNBS solution (1 mL). The solution was then incubated (50˚C/30 min in the dark). To stop the reaction, 0.1 M sodium sulphite (2.0 mL) was added. After cooling down to room temperature (RT; 27–29˚C) for 15 min, the absorbance was measured at 420 nm. L-leucine (0–1.5 mM) was used as standard. The AA content was then calculated and reported as mmol L-leucine equivalent/g protein. DH was estimated as follows (1):

$$DH\ (\%) = \left[\frac{L_t - L_0}{L_{max} - L_0}\right] \times 100 \tag{1}$$

where $L_t$ is the content of AA liberated at time t. $L_0$ is the content of AA in the original porcine placenta homogenate. $L_{max}$ is the total AA in the original porcine placenta homogenate obtained after acid hydrolysis (6 N HCl/95˚C/26 h). The acid-hydrolyzed placenta was filtered through Whatman paper no. 1 to remove the unhydrolyzed matters. Before AA quantification, the supernatant was neutralized with 6 N NaOH.

## Measurement of antioxidative activities

PPH obtained from different enzymatic hydrolysis conditions were measured for DPPH•/ ABTS•+ scavenging activities, ferric-reducing antioxidant power (FRAP), and metal chelating activity.

For the DPPH• inhibition, the method of Tkaczewska et al. [23] was used. Briefly, 300 μL of diluted PPH was mixed with 0.1 mM DPPH in 95% ethanol (1.5 mL). After thoroughly shaking and incubating in the dark for 30 min at RT, the absorbance was taken at 517 nm. Standard curve was made using Trolox (0–0.2 mM). DPPH• scavenging activity was reported as mmol Trolox equivalent (TE)/ g protein.

ABTS•+ inhibition was performed according to the method of Kim et al. [24]. The stock solutions composed of 7 mM ABTS•+ solution and 2.45 mM potassium persulphate solution. The working solution was made by combining the stock solutions at ratio of 1:1 (v/v). After standing for 12–16 h at RT in the dark, the working solution was diluted with 95% methanol to gain an absorbance of 0.70 ± 0.02 units at 734 nm using a UV-1900 Shimadzu spectrophotometer (Kyoto, Japan). ABTS•+ solution was newly prepared before analysis. In the reaction, 500 μL of diluted PPH was mixed with 2.5 mL of ABTS•+ solution and incubated at RT for 6 min in the dark. Then, the absorbance was read at 734 nm. Trolox (0–0.2 mM) was used as standard. ABTS•+ scavenging activity was reported as mM TE/ g protein.

FRAP was analyzed according to Tkaczewska et al. [23]. The FRAP reagent consisted of 10 mM TPTZ in 40 mM HCl (10 mL), 20 mM ferric chloride solution (10 mL), and acetate buffer, pH 3.6 (100 mL). PPH solution (100 μL) was added with the freshly prepared FRAP solution (1.9 mL). The mixture was incubated at RT in the dark for 10 min and the absorbance was measured at 595 nm. The standard curve of Trolox (0–0.16 M) was made and the results were reported as the mmol TE/g protein.

The method of Sripokar et al. [22] was used for determination of the $Fe^{2+}$ chelating activity. Sample (4.70 mL) was diluted with distilled water at the ratio of 1:1 (v/v). Then, 0.2 mM $FeCl_2$ (0.1 mL) and 0.50 mM ferrozine (0.2 mL) were added. The absorbance was measured at 562 nm after incubation at RT for 20 min. For control, the distilled water was used instead of the

sample. For blank, the distilled water was used instead of ferrozine. The metal chelating activity was estimated as follows (2):

$$\text{Metal chelating activity (\%)} = \left[1 - \left(\frac{\text{Abs}_{\text{sample}} - \text{Abs}_{\text{blank}}}{\text{Abs}_{\text{control}}}\right)\right] \times 100 \qquad (2)$$

## Antioxidative stability in the GI tract digestion model

The *in vitro* digestion tract model system was prepared as per Seiquer et al. [25]. Freeze-dried PPH (1 g) was added with 60 mL of distilled water and pepsin (0.05 g pepsin/g sample). Then, the pH was brought to 2.0 with 6 M HCl. The reaction mixture was incubated at 37°C in a shaking water bath (110 rpm) for 60 min. To terminate the reaction, 3 mL of the mixture was immediately placed to hot water (95°C) for 5 min. The solution was centrifuged at 9,000 ×g for 15 min and the supernatant was obtained. The remaining solution (30 mL) was placed in beaker and 6 N NaHCO$_3$ solution was added to neutralize the titratable acidity. Thereafter, 2.5 mL of the mixture of pancreatin and bile salts (0.1 g of pancreatin and 62.5 mL of bile salts in 0.1 NaHCO$_3$ solution) was immediately mixed. The pH of the mixture was brought to 7.5 with 6 N NaHCO$_3$. The sample solution was again kept at 37°C in a shaking water bath (110 rpm) for 300 min. Finally, the solution was taken, immediately incubated in hot water (95°C/5 min), and centrifuged (9,000 ×g/15 min). DPPH$^{\bullet}$ and ABTS$^{\bullet+}$ inhibitions, as well as FRAP, were measured in all digestas.

## Antibacterial activity

The microbial inhibitory activity of PPH was tested against *Staphylococcus aureus* and *Escherichia coli* according to the agar-well diffusion method as described by Verma et al. [26]. Distilled water was employed as a control. The zone of inhibition (mm) was reported.

## Molecular weight distribution of PPH

To evaluate the molecular weight distribution of PPH, the size exclusion chromatography (SEC) was performed according to Sae-Leaw et al. [27] using mixed standards with the molecular weight of 12.4 kDa (cytochrome c from horse heart), 29 kDa (carbonic anhydrase from bovine erythrocytes), 66 kDa (albumin, bovine serum), 150 kDa (alcohol dehydrogenase from yeast), and 200 kDa (β-amylase from sweet potato). Void volume was measured using blue dextran (2,000,000 Da). All samples (8.0 mg/mL) was diluted using a phosphate buffer saline (PBS, pH 7.4). Chromatographic separations of standards and samples (1,000 μL) were performed with a Sigma Andrich Superdex$^{\circledR}$ column (200 Increase 10/300 GL Cytiva, 28-9909-44, column L × I.D. 30 cm × 10 mm, 8.6 μm particle size). A low-pressure chromatography system (<1.3 mPa) and a fraction collector (Bio-Rad Laboratories, Hercules, CA) were used. For elution, PBS was used at a flow rate of 0.4 mL/min. Fractions (3 mL) were taken for absorbance measurement at 220 and 280 nm.

## Fourier transform infrared spectrometer (FTIR)

The freeze-dried PPH was ground and passed through a 100-mesh sieve according to Chen et al. [28]. PPH powder was compressed at 50 N into a circle slice and fixed into a slide holder. Scanning was performed at 4,000 cm$^{-1}$ to 530 cm$^{-1}$ using attenuated total reflectance (ATR); miracle ATR accessory (ZnSe single crystal) mode with a rate of 4 cm$^{-1}$/point at the average of 16 scans using a Bruker Model Vector 33 FTIR spectrometer (Bruker Co., Ettlingen, Germany) at RT. Spectral data was analyzed by the OPUS 3.0 data collection software program [29].

## Statistical analysis

All experiments were run in triplicate with three different lots of placenta (N = 3) and data were presented as mean ± standard deviation. A probability value $P < 0.05$ was considered significant. A mean comparison was done by Duncan's multiple range tests. In the case of pair comparison, a paired $t$-test was used. Statistical analysis was performed using SPSS for Windows Version 17.0 (SPSS Inc., Chicago, IL, USA).

The correlation coefficient between enzymatic hydrolysis parameters and antioxidative activities was also conducted using the SPSS. All samples were normal distribution and homogeneity of variance. Hydrolysis parameters including enzyme concentration, hydrolysis time, and DH on the relation to antioxidant assays (DPPH, ABTS, FRAP, and metal chelation) were analyzed by one-way ANOVA. The least square means were analyzed using Duncan's post hoc test using 95% significant level ($P < 0.05$). Correlation coefficient among antioxidants tests were analyzed using the Pearson's linear correlation.

## Results

### Chemical compositions of porcine placenta

The chemical compositions, including proximate composition, mineral profile, amino acid composition, and TCA-soluble peptide of freeze-dried porcine placenta are presented in Table 1. Protein was found to be the most abundant component in freeze-dried porcine placenta (78.03%), followed by ash (8.59%), moisture (6.61%), carbohydrate (3.34%), lipid (2.40%), and fiber (0.61%), respectively ($P < 0.05$). In freeze-dried porcine placenta, a TCA soluble peptide of 48.12 mmol tyrosine/g was found (Table 1), demonstrating the presence of endogenous peptides. Calcium was the most abundant element (1789.09 mg/100 g), followed by phosphorus (1052.54 mg/100 g), sodium (571.49 mg/100 g), and potassium (224.42 mg/100 g). The concentrations of other elements (iron, magnesium, copper, and manganese) ranged from 0.81–61.64 mg/100 g. (Table 1).

Twenty amino acids were detected in freeze-dried porcine placenta (Table 1). The top five most prevalent amino acids were glutamic acid, proline, glycine, aspartic acid, and leucine. All nine essential amino acids (EAA) were found in porcine placenta (Table 1), indicating that it could be a good source of high quality protein. Among EAA, leucine had the highest content (5.73 g/100 g), followed by lysine (4.05 g/100 g), valine (3.98 g/100 g), phenylalanine (2.76 g/100 g), threonine (2.70 g/100 g), histidine (1.82 g/100 g), isoleucine (1.78 g/100 g), methionine (1.07 g) (Table 1). Other non-essential amino acids (NEAA) were found in concentrations ranging from 0.62–9.73 g/100 g. The most abundant NEAA was glutamic acid, which was followed by proline, glycine, aspartic acid, hydroxyproline, alanine, arginine, serine, tyrosine, cysteine, and cysteine (Table 1).

### DH as affected by enzyme type, enzyme concentration, and reaction time

DH of PPH produced by Alcalase, Flavourzyme, and papain at various concentrations and reaction times are depicted in Fig 1a–1c. Overall, DH was gradually increased with increasing hydrolytic time in all samples ($P < 0.05$). At the end of hydrolysis (20 min), Alcalase with 10% enzyme load showed the maximum hydrolysis activity toward production of PPH with a DH of 36.55% (Fig 1a), whereas papain at the same concentration performed the slowest hydrolysis with a maximum DH of 29.82% (Fig 1c). When enzyme concentration was raised in all proteolysis reactions, an increase in DH was typically noticed. The hydrolytic rates, on the other hand, varied depending on the enzyme types and concentrations. The highest DH of all tested enzymes can be found at 20 min of hydrolysis, in which 10% Alcalase produced PPH with

**Table 1. Proximate composition, TCA-soluble peptide, mineral profile, and amino acid composition of freeze-dried porcine placenta.**

| Composition | Contents |
|---|---|
| Proximate composition (%) | |
| Moisture | 6.61±0.03 |
| Protein | 78.03±0.30 |
| Fat | 2.40±0.40 |
| Crude fiber | 0.61±0.01 |
| Ash | 8.59±0.99 |
| Carbohydrate | 3.34±0.16 |
| TCA-soluble peptide (mmol tyrosine/g) | 48.12±0.77 |
| Minerals (mg/100 g) | |
| Calcium | 1789.09±51.58 |
| Phosphorus | 1052.54±29.01 |
| Sodium | 571.49±11.89 |
| Potassium | 224.42±0.79 |
| Magnesium | 61.64±1.23 |
| Iron | 20.76±0.20 |
| Zinc | 5.54±0.10 |
| Manganese | 0.81±0.03 |
| Copper | 0.84±0.01 |
| Amino acid composition (g/100 g) | |
| Essential amino acid (EAA) | |
| Leucine | 5.73±0.23 |
| Lysine | 4.05±0.33 |
| Valine | 3.98±0.25 |
| Phenylalanine | 2.76±0.08 |
| Threonine | 2.70±0.06 |
| Histidine | 1.82±0.20 |
| Isoleucine | 1.78±0.06 |
| Methionine | 1.07±0.04 |
| Tryptophan | 0.71±0.01 |
| Non-essential amino acid (NEAA) | |
| Glutamic acid | 9.73±0.33 |
| Proline | 8.70±0.17 |
| Glycine | 8.49±0.29 |
| Aspartic acid | 6.85±0.12 |
| Alanine | 4.80±0.19 |
| Arginine | 4.48±0.15 |
| Hydroxyproline | 3.89±0.46 |
| Serine | 3.67±0.15 |
| Tyrosine | 2.09±0.17 |
| Cysteine | 0.91±0.02 |
| Cystine | 0.62±0.01 |

Values are given as mean ± standard deviation from triplicate determinations.

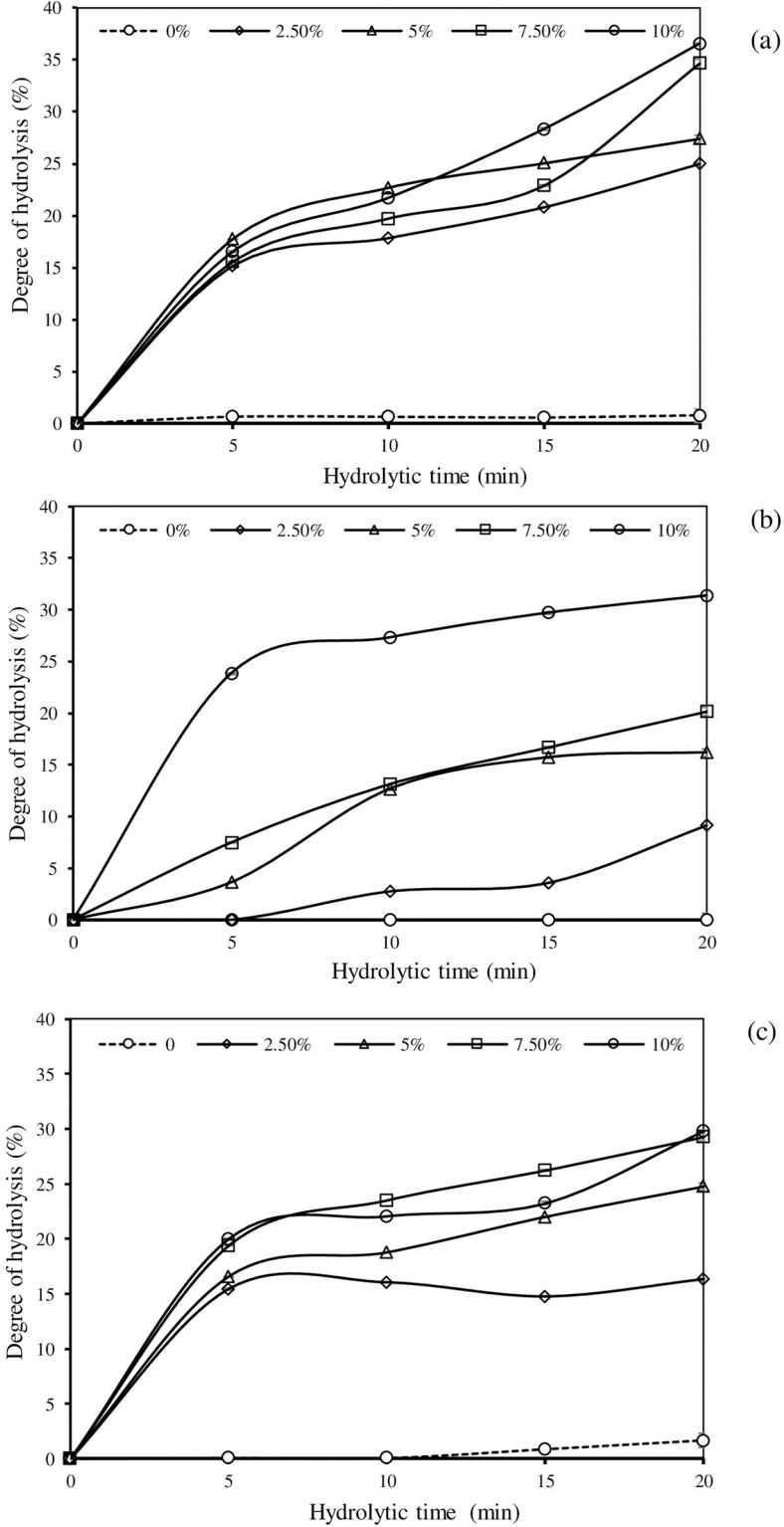

**Fig 1. Effects of enzyme type (Alcalase (a), Flavourzyme (b), and papain (c)), enzyme concentration, and hydrolysis time on the degree of hydrolysis (DH) of porcine placenta protein.** Bars represent standard deviation from triplicate determinations.

36.55% DH (Fig 1a), 10% Flavourzyme gave 31.37% DH (Fig 1b), and 7.50% and 10% papain provided 29.82% DH (Fig 1c) ($P < 0.05$). During the production of PPH, Alcalase and Flavozyme showed higher hydrolytic activity than papain (Fig 1a–1c). It's also worth noting that the hydrolysis rates markedly increased for the first 5–10 min before gradually increasing (Fig 1a–1c).

## Antioxidant activities of PPH as affected by enzyme type, enzyme concentration, and hydrolysis time

The DPPH$^\bullet$ and ABTS$^{\bullet+}$ scavenging activities, metal chelation, and reducing activity of PPH obtained by different proteolysis enzymes, enzyme concentrations (0, 2.5, 5, 7.5, and 10%) and hydrolysis times (0, 5, 10, 15, and 20 min) were investigated (Fig 2). All PPH had stronger DPPH$^\bullet$ scavenging activity than the parent porcine placenta (Fig 2a–2c). Fresh porcine placenta (without enzyme) shows a slight increase in DPPH$^\bullet$ inhibition with time, indicating that endogenous proteases were involved in the proteolysis process (Fig 2a–2c). The DPPH$^\bullet$ scavenging activity of PPH tended to increase with increasing enzyme concentration and proteolysis time, indicating that smaller peptides have higher DPPH$^\bullet$ scavenging activity ($P < 0.05$). At the same enzyme concentration and reaction time, PPH produced by papain had a higher DPPH$^\bullet$ scavenging activity than PPH produced by other enzymes ($P < 0.05$). PPH catalyzed by 10% papain for 20 min had the strongest DPPH$^\bullet$ inhibition, ranging from 120.57 to 284.66 mmol TE/g protein ($P < 0.05$). In summary, the enzyme type, enzyme load, and hydrolysis time all influenced PPH's DPPH$^\bullet$ scavenging activity.

The ABTS$^{\bullet+}$ scavenging activity of PPH was boosted by increasing enzyme load and incubation time (Fig 2d–2f), which was similar to the DPPH$^\bullet$ inhibitory effect. The ABTS$^{\bullet+}$ scavenging activity of the original fresh placenta was 111.74 mmol TE/g protein, which gradually increased after enzymatic hydrolysis (Fig 2d–2f). When comparing PPH catalyzed by Flavourzyme to PPH catalyzed by other enzymes at the same enzyme concentration and reaction duration, PPH catalyzed by Flavourzyme demonstrated exceptional ABTS$^{\bullet+}$ scavenging activity (Fig 2e). At 20 min, the maximum ABTS$^{\bullet+}$ inhibition was found in 10% Flavourzyme produced PPH, with 5,641.73 mmol TE/g placenta protein ($P < 0.05$).

The reducing power of PPH produced by three different enzymes are depicted in Fig 3a–3c. The reducing power of PPH increased with increasing enzyme load and incubation time in general. PPH generated with 10% enzyme at 20 min had the maximum reducing capacity, in which Alcalase rendered the PPH with the greatest reducing value (85.07 mmol TE/g), followed by Flavozyme (82.78 mmol TE/g), and papain (18.94 mmol TE/g) (Fig 3a–3c). This indicated that low-molecular-weight peptides with high DH had a higher reducing capacity than high-molecular-weight peptides. PPH created by mixed proteases (Alcalase and Flavourzyme) has roughly 4-fold stronger reducing ability than papain-produced PPH. As the enzyme concentration and hydrolytic time increased, so did the metal chelating activity (Fig 3d–3f). The smaller the peptide, the greater the metal chelating activity, according to this finding. It should be noted that PPH generated by Alcalase and Flavourzyme had higher metal chelating capacity than papain ($P < 0.05$), demonstrating that enzyme type has an effect on PPH chelating capacity. Overall, Alcalase-catalyzed PPH had the best metal chelating capability, particularly at high enzyme loads and extended hydrolytic times ($P < 0.05$).

The correlations between variations (enzyme load, hydrolytic time, and DH) and antioxidative properties of PPH were determined by Pearson's correlation analysis and the correlation coefficients ($r^2$) were reported. From Table 2, the enzyme concentration and DH were highly correlated with DPPH$^\bullet$ scavenging activity ($r^2 = 0.7517$–$0.9084$ for enzyme concentration; $r^2 = 0.8482$–$0.9563$ for DH), ABTS$^{\bullet+}$ scavenging activity ($r^2 = 0.7436$–$0.9462$ for enzyme

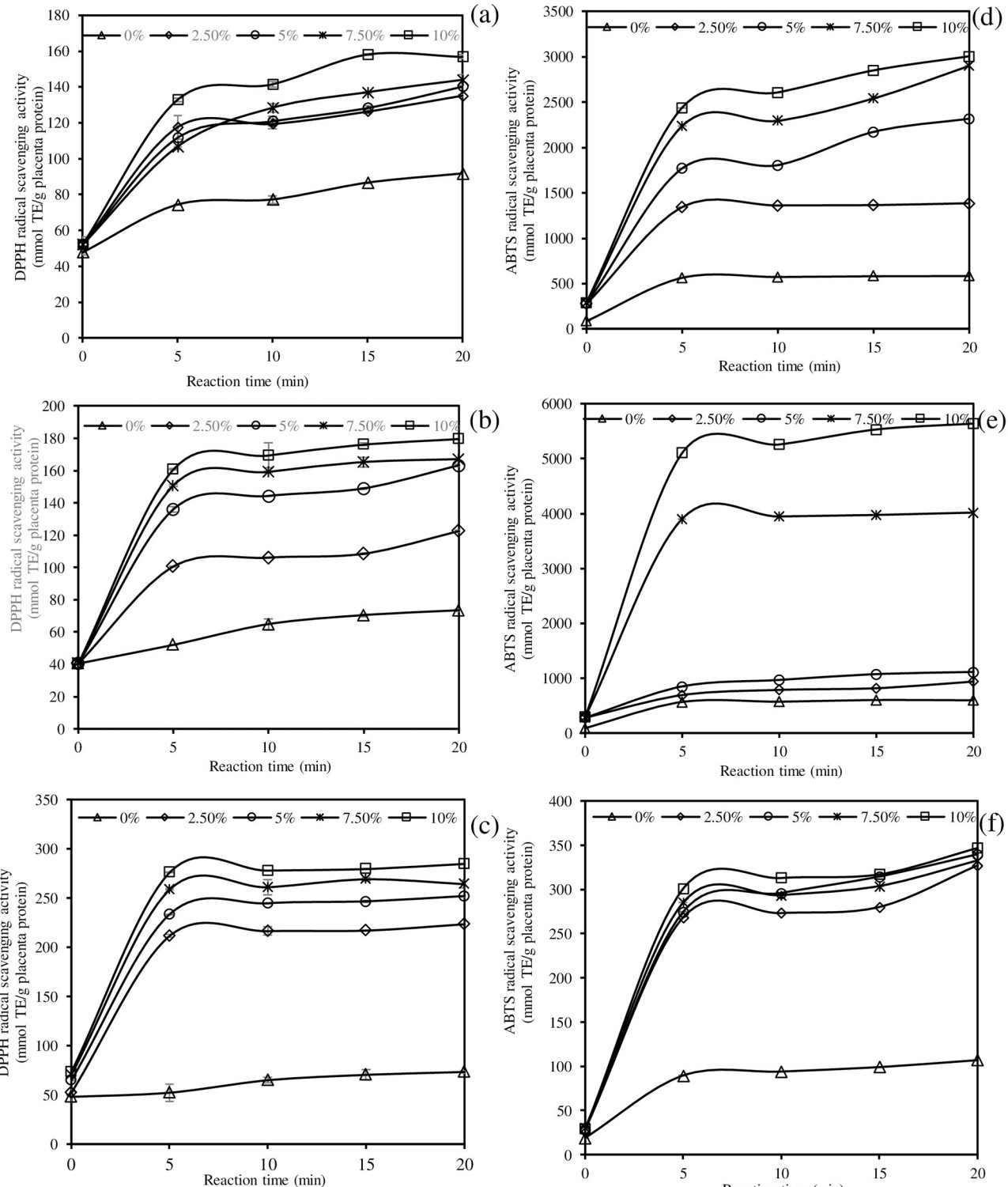

**Fig 2. Effects of enzyme type (Alcalase (a, d), Flavourzyme (b, e), and papain (c, f)), enzyme concentration, and hydrolysis time on the DPPH•** **scavenging activity (a, b, c) and ABTS•+ scavenging activity (d, e, f) of porcine placenta hydrolysate (PPH).** Bars represent standard deviation from triplicate determinations. TE = trolox equivalent.

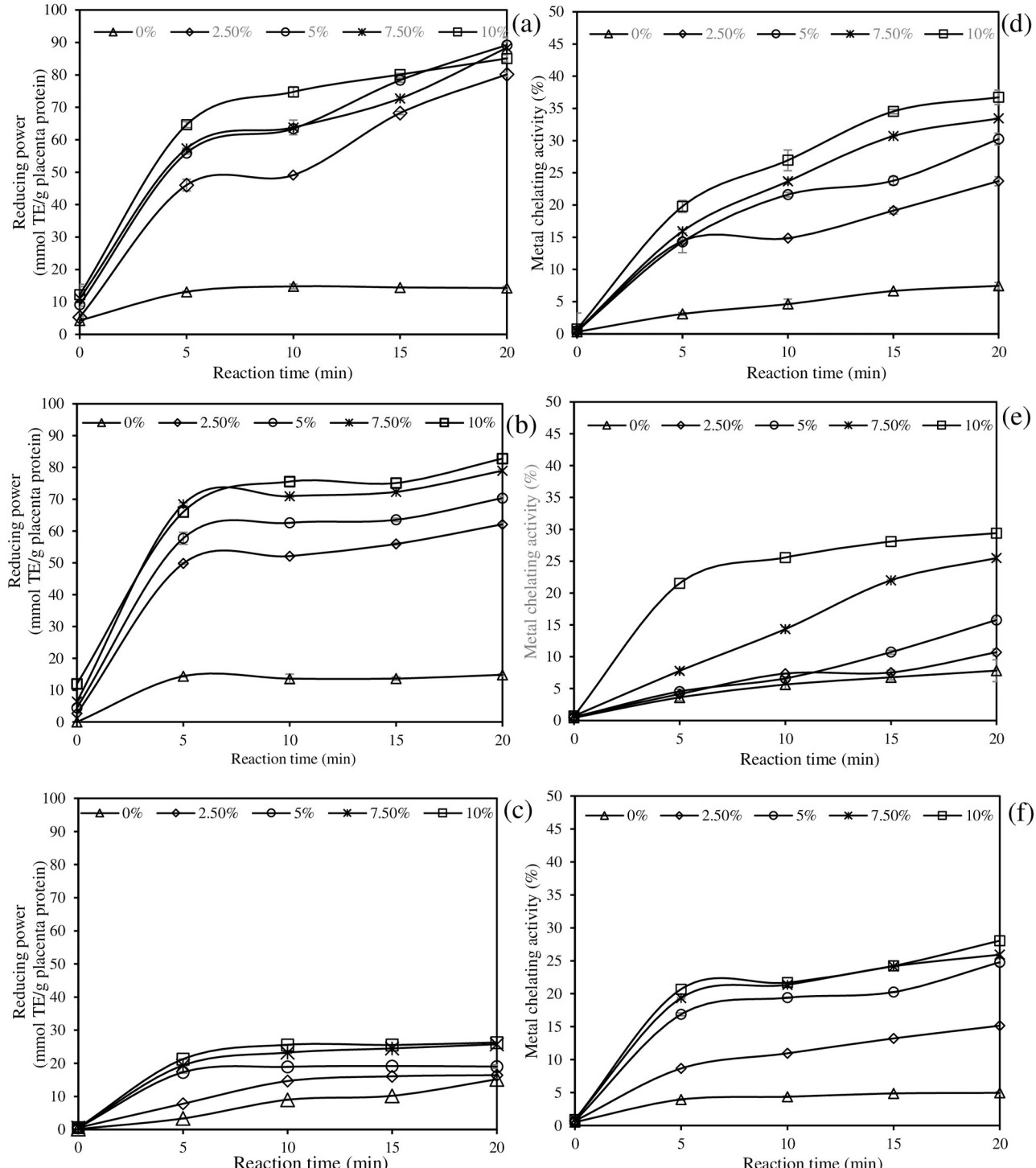

**Fig 3. Effects of enzyme type (Alcalase (a, d), Flavourzyme (b, e), and papain (c, f)), enzyme concentration, and hydrolysis time on the reducing power (a, b, c) and metal chelating activity (d, e, f) of porcine placenta hydrolysate (PPH).** Bars represent standard deviation from triplicate determinations.

**Table 2. Correlation coefficient of enzymatic hydrolytic variables and antioxidant activities of porcine placenta hydrolysate.**

| Enzyme | Variable | DPPH˙ scavenging activity | ABTS˙⁺ scavenging activity | Metal chelating capacity | Reducing power |
|---|---|---|---|---|---|
| Alcalase | Hydrolysis time | 0.4035 | 0.2059 | 0.5050 | 0.3817 |
| | Enzyme load | 0.7517 | 0.9462 | 0.7512 | 0.7583 |
| | DH | 0.9400 | 0.8308 | 0.9496 | 0.9362 |
| Flavourzyme | Hydrolysis time | 0.3480 | 0.0619 | 0.4417 | 0.2059 |
| | Enzyme load | 0.8757 | 0.9166 | 0.7522 | 0.8647 |
| | DH | 0.8482 | 0.7181 | 0.9301 | 0.7038 |
| Papain | Hydrolysis time | 0.0651 | 0.1514 | 0.3044 | 0.1622 |
| | Enzyme load | 0.9084 | 0.7436 | 0.8854 | 0.9515 |
| | DH | 0.9563 | 0.9205 | 0.9650 | 0.8830 |

DH = degree of hydrolysis.

concentration; $r^2$ = 0.7181–0.9205 for DH), metal chelating capacity ($r^2$ = 0.7512–0.8854 for enzyme concentration; $r^2$ = 0.9301–0.9650 for DH), and reducing activity ($r^2$ = 0.7583–0.9515 for enzyme concentration; $r^2$ = 0.7038–0.9362 for DH). However, there was a weak correlation between hydrolysis duration and antioxidative effects, particularly in papain-derived PPH ($r^2$ = 0.0651–0.3044). The DH was the most powerful enzymatic proteolysis parameter that influenced the antioxidant capabilities of PPH, with the greatest $r^2$ value in all samples ($P < 0.05$).

## Antioxidative stability in GI tract model system

In comparison to fresh placenta (control), changes in antioxidative activities of PPH generated by three proteases at 10% enzyme concentration for 20 min after digestion through GI tract model system are shown in Fig 4. Fresh placenta and PPH were separated into two fractions: gastric and gastrointestine extracts, which mimicked the digesta in the stomach and intestine, respectively. Fresh placenta's DPPH˙ scavenging activity, as well as all three PPH, rose in the stomach condition and thereafter decreased in the intestine condition ($P < 0.05$, Fig 4a). The ABTS˙⁺ scavenging activity of all samples, on the other hand, tended to diminish in stomach condition before increasing in intestinal condition (Fig 4b). FRAP of PPH derived from three different enzymatic hydrolysis showed similar tendencies to DPPH˙ scavenging activity. Under stomach condition, FRAP increased in all PPH, while it reduced slightly under intestinal environment (Fig 4c). When PPH was transported via the GI tract model system, its antioxidative actions were significantly altered.

## Antibacterial activity of PPH

Table 3 shows the antibacterial activity of PPH produced with three different enzymes at a 10% enzyme concentration for 20 min against *S. aureus* and *E. coli*. The zone of inhibition was used to determine the extent of PPH's antibacterial activity. Depending on the enzyme used, fresh placenta and PPH demonstrated antibacterial activity against *S. aureus* and *E. coli* to varying degrees. The bactericidal activity of fresh placenta and Flavourzyme-made PPH against *S. aureus* and *E. coli* were the lowest ($P < 0.05$). PPH derived from papain had the greatest inhibitory activity against *S.aureus* and *E.coli*, with antibacterial activity 4.65 and 4.33 times that of fresh placenta, respectively ($P < 0.05$). The PPH produced by Alcalase inhibited both bacteria in the second order (Table 3), with inhibition of *S. aureus* and *E. coli* being 1.53 and 2.78 times higher than fresh placenta, respectively. It's worth noting that fresh placenta and its PPH are more efficient at inhibiting gram-positive *S. aureus* than gram-negative *E. coli*.

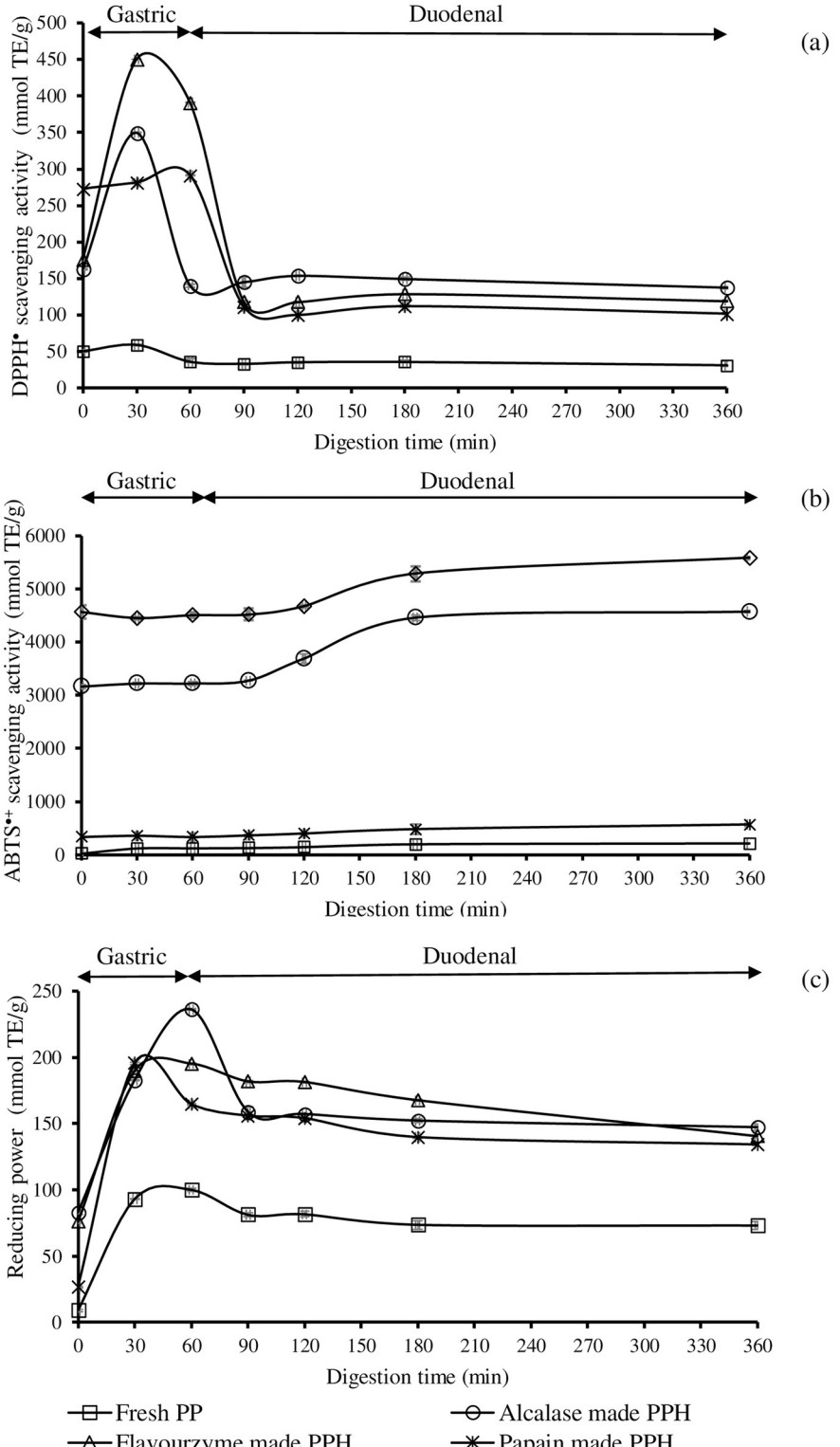

**Fig 4. Changes in DPPH• scavenging activity (a), ABTS•+ scavenging activity (b), and reducing power (c) of porcine placenta hydrolysate (PPH), produced by Alcalase, Flavourzyme, and papain at 10% enzyme load for 20 min, in gastrointestinal (GI) tract model system.** To compare, fresh porcine placenta (Fresh PP) was employed. Bars represent standard deviation from triplicate determinations. TE = trolox equivalent.

**Table 3. Antibacterial activity (zone of inhibition) of porcine placenta hydrolysate (PPH) catalyzed by three different enzymes at 10% enzyme concentration for 20 min.**

| Sample | Clear zone (mm) | |
|---|---|---|
| | *S.aureus* | *E.coli* |
| Fresh placenta | 2.30±0.78[a] | 1.80±0.00[a] |
| Alcalase made PPH | 3.52±0.19[b] | 5.00±0.38[b] |
| Flavourzyme made PPH | 2.50±0.45[a] | 2.80±0.94[a] |
| Papain made PPH | 10.70±0.17[c] | 7.80±0.74[c] |

Values are given as mean ± standard deviation from six determinations.

Different letters in the same row indicate significant differences ($P < 0.05$).

## Molecular weight distribution of PPH

In comparison to fresh porcine placenta, the molecular weight distribution of PPH produced by three enzymes at a 10% enzyme concentration for 20 min is shown in Fig 5. In comparison to the original fresh placenta, there was a reduction in protein molecular mass in PPH. A major protein with a molecular mass of 338.13 kDa was found in the fresh porcine placenta. Low molecular peptides were found in PPH after hydrolysis, indicating that three proteases created smaller peptides. Six major protein fractions with differing molecular weights were discovered in fresh placenta and its hydrolysates using size exclusion chromatography. Fresh porcine placenta protein had a molecular mass of 12.34–338.13 kDa, which was reduced to smaller molecules after proteolysis. Among enzyme, Alcalase can create the smallest peptides, with molecular weights ranging from 0.46 to 303.5 kDa. Alcalase's PPH was made up of three small peptide fractions with molecular masses of 6.50, 1.76, and 0.46 kDa. The PPH generated by Flavourzyme and papain had a similar peptide size. Papain-produced PPH, on the other hand, had two small peptide fractions with molecular masses of 1.78 and 0.40 kDa.

## FTIR spectra

The characteristic FTIR bands represented the proteolysis such as the amide I (1,700–1,600 cm$^{-1}$) and II (1,590–1,520 cm$^{-1}$), the N-terminal (NH$^{3+}$, 1,510 cm$^{-1}$) and the C-terminal (COO$^{-}$, 1,400 cm$^{-1}$) were detected in all PPH (Fig 6). The NH$^{3+}$ (~1516 cm$^{-1}$) and COO$^{-}$ (~1400 cm$^{-1}$) bands in all PPH were clearly visible compared to the fresh porcine placenta. When comparing PPH to fresh placenta, the FTIR bands associated with distinct peptide conformations at below 1,400 cm$^{-1}$ (amide III) were likewise observed in all PPH (Fig 6), indicating that proteolysis had occurred. Fresh porcine placenta had bands around 2,800–3,500 cm$^{-1}$ that were clearly evident, although these bands appeared to be diminished after PPH production.

## Discussion

### Chemical compositions of porcine placenta

Previous researches demonstrated the nutrient-rich source of placenta including proteins and peptides [30]. All essential nutrients for fetus development are transported to fetus via placenta [31]. Protein was the most abundant proximate composition of freeze-dried porcine placenta (~77%), which was in agreement with Jang et al. [32] who reported 78% crude protein in porcine placenta. Collagenous proteins are presented in the animal placenta as indicated by the presence of hydroxyproline (Table 1). Hydroxyproline is the amino acid characteristic of collagen [33]. About 10% of ash was detected in porcine placenta which was possibly associated

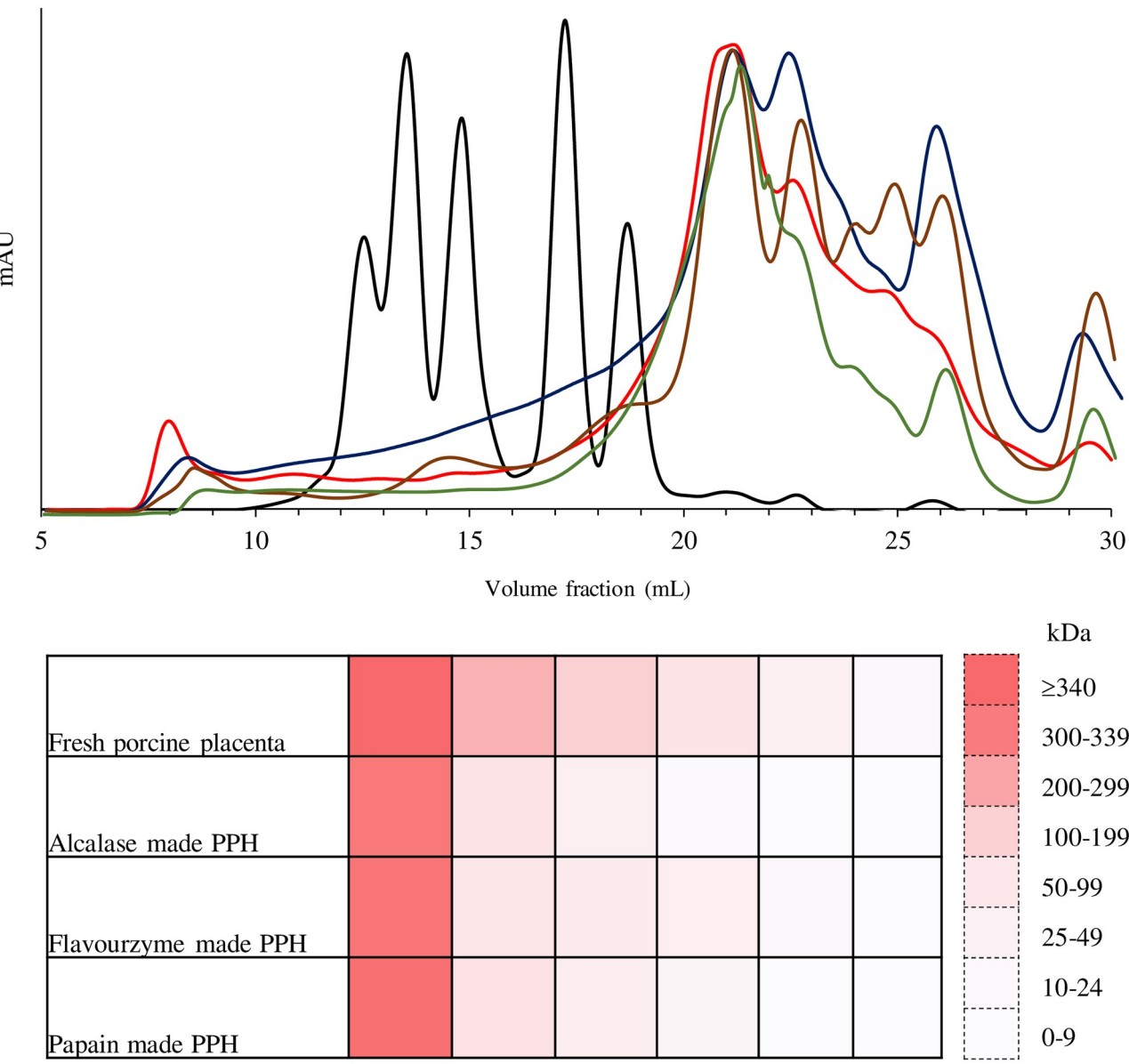

**Fig 5. Molecular weight distribution of fresh porcine placenta (brown line) and porcine placenta hydrolysate (PPH) produced by Alcalase (red line), Flavourzyme (blue line), and papain (green line) at 10% enzyme load for 20 min.** Standard protein markers are represented by the black peaks.

with the minerals required for fetus development [31]. This value was higher than the report of Jang [32] in freeze-dried porcine placenta, which accounted for 5.94%. Other components of freeze-dried placenta, such as fat, carbohydrate, and fiber, were detected in low levels. According to Hay [31], glucose and fatty acids were transferred into the placenta by facilitative diffusion and direct transporter-mediated fatty acid transfer or uptake from lipoprotein.

Herein, the main minerals found in porcine placenta were calcium and phosphorus. The findings contradicted those of Jang et al. [32], who found sodium and phosphorus to be the most abundant components in freeze-dried porcine placenta, followed by calcium and potassium. The iron concentration of the porcine placenta in this study (207.60 mg/kg) was lower than in a previous report (1238.12 mg/kg) [32]. Fresh porcine placenta were washed twice with

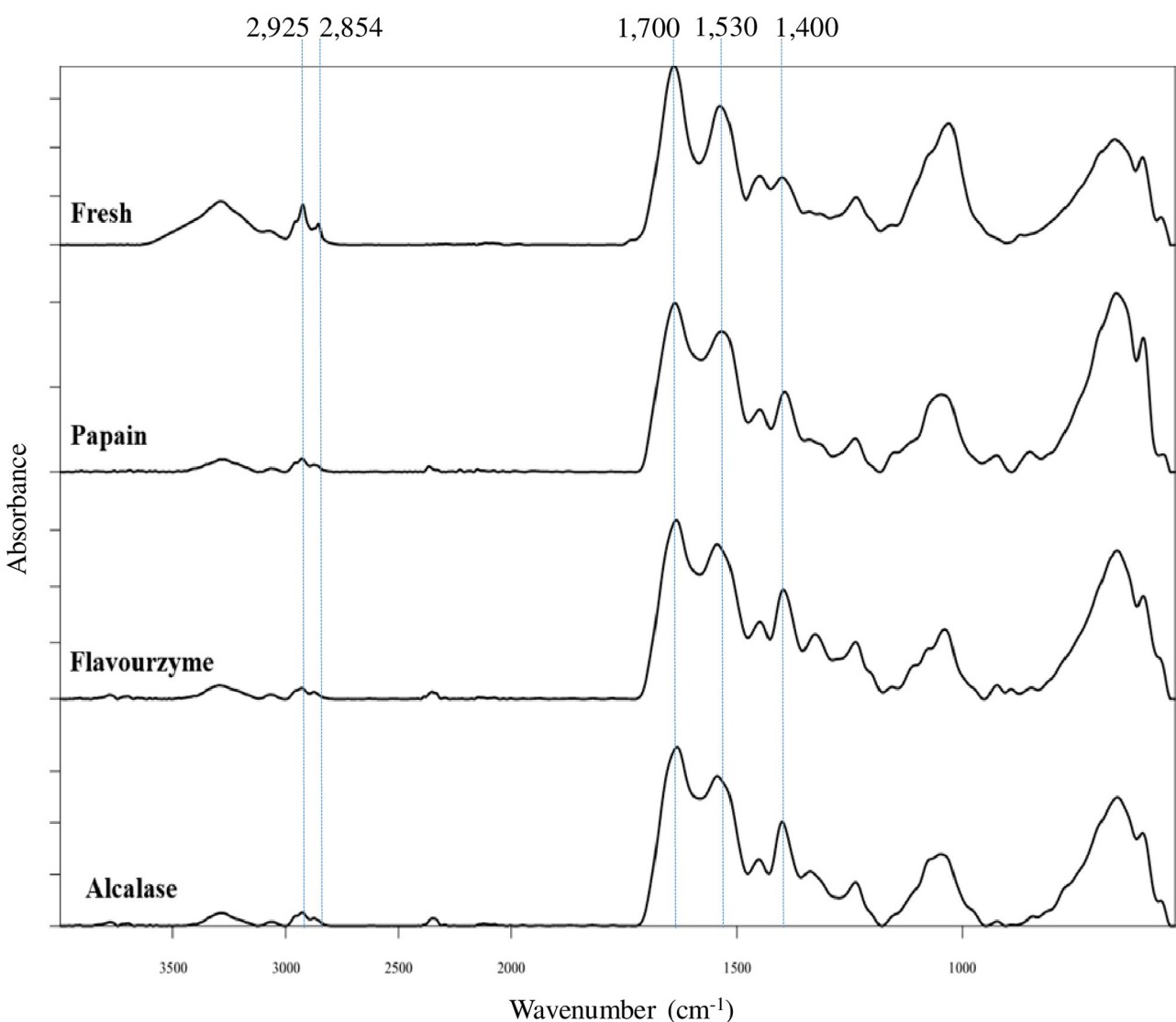

2,925  2,854          1,700  1,530  1,400

**Fig 6. Fourier transform infrared (FTIR) spectra of freeze-dried porcine placenta (fresh) and freeze-dried porcine placenta hydrolysate (PPH) produced by Alcalase, Flavourzyme, and papain at 10% enzyme load for 20 min.**

cold water to remove blood and mucus before being freeze-dried in this investigation. When blood was removed, iron-containing sarcoplasmic proteins, particularly hemoglobin, were also removed [34]. Minor elements such as zinc and manganese were also found in porcine placenta. The variation in individual mineral content between this study and the prior report could be attributed to variances in species, feed ingredients, and pre-treatment (e.g. washing procedure). The presence of TCA-soluble peptides suggested the presence of endogenous and exogenous peptides formed during post-harvest processing by autolysis of porcine placenta. The placenta contains endogenous peptides such as oxytocin (nonapeptide), glycylsarcosine (dipeptide), and opioid peptides [30].

The porcine placenta had twenty amino acids containing all nine EEA. The highest EEA was leucine (Table 1), which matched the findings of Jang et al. [32] in freeze-dried placenta. Glutamic, proline, and glycine were the most abundant amino acids in porcine placenta, while

cysteine and tryptophan were rare (Table 2). Results were also in agreement with Jang et al. [22] who reported that porcine placenta predominantly composed of glutamic acid, proline and glycine with low levels of cysteine and tryptophan. The presence of hydroxyproline in the porcine placenta indicates the existence of collagen. Typically, hydroxyproline is a unique amino acid presented in collagen [35]. Interestingly, the hydrophobic amino acids isoleucine, leucine, methionine, phenylalanine, proline, tryptophan, and valine were found in abundance in porcine placenta. These hydrophobic amino acids promoted the functionality and bioactivity of food-derived bioactive peptides containing 2–20 amino acid residues in a synergistic manner [36]. Cheung et al. [37] and Majumder and Wu [38] suggested that the occurrence of aromatic amino acid residue at the C-terminus and hydrophobic amino acid residue at the N-terminus of peptide enhanced the angiotensin I-converting enzyme (ACE) inhibitory activity. In addition, an improvement of bioavailability and antioxidative properties of casein-derived peptides in simulated GI fluids was noticed in high hydrophobicity peptides [39]. Thus, porcine placenta is regarded as a potential substrate for protein hydrolysate preparation due to its high protein content and proper amino acid composition.

## DH as affected by enzyme types and concentrations

The degree of proteolysis influences the bioactivities and functionalities of protein hydrolysates, which are determined by DH [40]. Increasing enzyme load and hydrolytic time resulted in increasing DH of PPH (Fig 1), depending on the enzyme class. Noman et al. [41] suggested that the longer the incubation time, the more extensively protease acted on the protein molecule, resulting in an increasing DH. Among the enzymes, Alcalase catalyzed the highest hydrolysis rate (Fig 1a), with the highest DH when compared at the same hydrolysis time ($P < 0.05$). Alcalase preferentially hydrolyzes hydrophobic amino acid residues [42], resulting in a greater extent of hydrolysis of porcine placenta. Overall, an increase in DH of PPH was observed after 20 min of hydrolysis. However, the hydrolysis rates varied depending on the enzyme used (Fig 1). The higher degree of hydrolysis was observed within the first 5 min, followed by a slower degree up until the end of the process (20 min). A faster rate of proteolysis resulted in more peptide bond cleavage, whereas a slower rate may be associated with decreased enzyme activity due to fewer peptide bonds remaining or a product inhibited reaction [42]. With the same enzyme concentration and hydrolysis time, a higher DH was observed in PPH made by Alcalase, indicating that porcine placenta is more susceptible to Alcalase catalyzed reactions. According to Hamzeh et al. [43], the presence of proline and/or hydroxyproline in splendid squid gelatin contributed to the lower DH of its hydrolysate caused by proteolysis resistance.

## Antioxidative activities of PPH as affected by enzyme type, enzyme concentration, and hydrolysis time

Without enzyme, porcine placenta revealed some DPPH• scavenging activity, indicating the presence of some endogenous antioxidant peptides and free amino acids. When compared to other enzymes at the same enzyme concentration and hydrolysis time, the PPH produced by papain demonstrated excellent DPPH• scavenging activity (Fig 2c). Papain prefers to cleave peptides with hydrophobic units (alanine, valine, leucine, isoleucine, phenylalanine, tryptophan, and tyrosine) after arginine or lysine but not after valine [44]. As a result, the peptide sequence of PPH produced by papain may contain a high concentration of hydrophobic amino acids, which may scavenge the hydrophobic DPPH•. With increasing DH of all PPH, an increase in DPPH• scavenging activity was observed. Smaller bioactive peptides may more easily react with free radicals and convert them to stable products. The antioxidant activity of porcine liver hydrolysates catalyzed by Alcalase, Flavourzyme, and papain was also DH-

dependent [26,45]. The capacity for free radical binding was increased at high DH. This result agreed with Hamzeh et al. [43] and You et al. [46] on DPPH• inhibition of squid gelatin hydrolysate (DH = 10–50%) catalyzed by crude protease from Pacific white shrimp hepatopancreas and loach protein hydrolysate catalyzed by Alcalase and papain (DH = 18–23%). Antioxidative peptides of various sizes and amino acid compositions/sequences can be generated during enzyme-catalyzed protein hydrolysis depending on enzyme specificity [47]. However, after 5 min of hydrolysis, there was a slight increase in the DPPH• scavenging activities of PPH produced by papain at all concentrations (Fig 2c).

Fresh porcine placenta demonstrated some ABTS•+ scavenging activity (Fig 2d–2f), which could be attributed to intrinsic bioactive components such as peptides, free amino acids, nucleic acid, hormone, and α-tocopherol [32]. The increment of ABTS•+ scavenging activity in control placenta (without enzyme addition) during incubation for 20 min was most likely due to the activation of those intrinsic bioactive components as well as the formation of bioactive peptides produced by endogenous proteases of the original placenta (Fig 1a–1c). All PPH scavenged water soluble ABTS•+ more effectively than hydrophobic DPPH•, indicating that those PPH have a promising efficiency in scavenging the hydrophilic free radical. All PPH showed similar trends in DPPH• inhibitory activity and ABTS•+ scavenging activity. Except for papain-made PPH, these antioxidant activities increased with increasing enzyme load and hydrolytic time (Fig 2d–2f). The increase in enzyme concentration had no effect on the ABTS•+ scavenging activity of papain-derived PPH (Fig 2f). This PPH also had the lowest ABTS•+ inhibition when compared to other enzyme-produced PPH ($P < 0.05$). The highest ABTS•+ inhibition was found in PPH made by 10% Flavozyme for 20 min ($P < 0.05$), which was 1.65 and 16.14 folds higher than PPH made by Alcalase and papain, respectively, when compared to similar enzyme concentration and hydrolysis time. The increased ABTS•+ scavenging activity was most likely due to the presence of a number of hydrophobic residues [48]. According to the amino acid composition of porcine placenta, the high content of hydrophobic amino acids, specifically glycine, proline, alanine, valine, and leucine (Table 1), could potentially transfer electron/hydrogen atoms to free radicals. Additionally, the amino acid composition and sequence of peptides can affect antioxidant activity [49]. As a result, the radical inhibitory activity of protein hydrolysate with the same DH produced by different enzymes can vary. Babini et al. [50] proposed that the presence of tryptophan/tyrosine at the C-terminus of peptides was associated with high ABTS•+ inhibition; however, amino acid sequences had less impact on DPPH• scavenging activity. The presence of hydrophobic amino acid residues leucine or valine at the N-terminus, as well as proline, histidine, or tyrosine in the sequence, are typical structural features of potent antioxidant peptides [51]. Hydrophobic amino acids found in peptides, such as tryptophane, phenylalanine, valine, isoleucine, glycine, lysine, and proline, have antioxidant properties [52]. Tian et al. [53] proposed that the antioxidant activity of peptides was due to tryptophan residues. As a result, Flavouzyme-produced peptides may contain the proper amino acid sequence, which can dramatically improve hydrogen atom or electron transfers.

During the first 5 min of proteolysis, PPH made by Alcalase and Flavourzyme demonstrated sharp increases in reducing power and metal chelating activity with increasing enzyme concentration ($P < 0.05$). Then, there was a slight increase in both activities. When compared to extensive hydrolysis, the results showed that limited hydrolysis produced PPH with better reducing power and metal chelation. This was consistent with the findings of You et al. [46] in loach protein hydrolysates with varying DH. The reducing power of PPH produced by papain was found to be lower, with a negligible change over the proteolysis time-course (Fig 3c). At high DH, Alcalase produced PPH with excellent reducing power and metal chelating activity (Fig 3a and 3d). This could be related to the amino acid composition of porcine placenta, polar

and non-polar amino acids (hydrophobicity), and protein hydrolysate side chains [45]. Tang et al. [7] found that the goat placenta protein extract with the highest reducing power had a molecular weight of 3–10 kDa and high levels of glutamic acid, glycine, aspartic acid, and tyrosine. The high content of polar amino acids (e.g., glutamic acid and aspartic acid) in porcine placenta (Table 1) may explain the PPH's superior metal chelating ability. However, the metal chelation of PPH was governed by the enzyme employed. Overall, the higher the enzyme concentration, the shorter the peptides chain. The hydrophobic amino acids had a chance to be exposed at that point. As a result, peptides with a higher hydrophobic residue content could perfectly bind to hydrophobic peroxyl radicals. The antioxidant activity of PPH was determined by the enzyme type, the time of hydrolysis, and the DH. Alcalase cleaves the peptide bond at the serine position, whereas papain breaks down the peptide bond at the cysteine position [45]. Flavourzyme is a broad specific enzyme [54]. As a result, peptides derived from three different enzymes had different amino acid sequences and chain lengths, suggesting that they may behave differently in terms of antioxidative potential and mode of action. PPH produced by papain demonstrated excellent DPPH• scavenging activity, whereas PPH produced by Flavourzyme demonstrated superior ABTS•+ scavenging activity, which may be related to the molecular mass of the peptides produced. Chai et al. [55] recommended that small collagen peptides with 1–2 kDa were the most powerful functional food or cosmetic ingredients. Furthermore, Alcalase produced PPH with higher reducing power and metal chelation ability.

For the relationship between the proteolytic parameters and the antioxidative properties of enzymatic PPH, the greater $r^2$ between the DH and all antioxidantive properties were found. Smaller peptides may be able to effectively react with free radicals and metal ions. The results showed that the DH had a strong influence on the antioxidant activities of PPH. Thammasena and Liu [56] discovered that the DH correlated positively with the antioxidant capacity of desalted duck egg white hydrolysate.

## Antioxidative stability in GI tract model system

The GI tract model systems simulate aspects of the human digestive system. The bioactivities of protein hydrolysate are determined by whether they are imperishable or stable against digestive enzymes, resulting in easy absorption in the human body. Increased DPPH• scavenging activity of all PPH after pepsin digestion (mimic stomach) could be attributed to increased exposure of hydrophobic side chains [57]. Hydrophobic DPPH• could easily access the exposed hydrophobic regions of PPH. After pancreatin digestion (mimic intestine), the DPPH• scavenging activity of PPH was dramatically reduced (Fig 4a). This could be due to the rapid accumulation of free amino acids and shorter peptides (di- and tri-peptides) caused by the excessive hydrolysis. As a result, more polar amino acids and shorter peptides struggled to react with the hydrophobic DPPH• [57]. This finding was also consistent with You et al. [46], who discovered a significant increase in DPPH• scavenging activity after pepsin digestion and a decrease after pancreatin digestion in loach peptides. The reduction of ABTS•+ scavenging activity of PPH after pepsin digestion was observed in an inverse fashion to the DPPH• scavenging activity. The increase trend was then observed after pancreatin digestion (Fig 4b), indicating that hydrophilic antioxidative peptides were produced in greater quantities during the simulated GI digestion. The disparity in ABTS•+ inhibition and DPPH• inhibition of GI digestas may have contributed to the increased hydrophobic peptides after pepsin digestion, which were unable to react with polar ABTS•+ properly. The increase in existing hydrophilic peptides after pancreatin digestion aided in their inhibition of ABTS•+. It was in agreement with You et al. [46] in loach peptides and Zhu et al. [57] in Alcalase-produced zein hydrolysate during a two-stage *in vitro* digestion.

The same trend was observed for reducing capacity as was observed for the DPPH result. It was proposed that existing small peptides after pepsin treatment could potentially reduce the $Fe^{3+}$/FeSCN complex to $Fe^{2+}$ form. It was consistent with the findings of Zhu et al. [57], who discovered an obviously increased reducing power of zein hydrolysate after 1 h of pepsin digestion. According to Megas et al. [58], the availability of bioactive components was increased after passing through the GI tract system. However, the antioxidative capacity and mode of action of peptides can vary depending on their structure and polarity.

## Antibacterial activities of porcine placenta hydrolysates

PPHs, particularly papain-derived PPH, demonstrated significant potential in *S. aureus* and *E. coli* inactivation (Table 3). The type of enzyme used determined PPH's antibacterial activity. PPH derived from papain demonstrated excellent antibacterial activity against *S. aureus* and *E. coli* by forming the largest zone of inhibition ($P < 0.05$). The peptide sequence and chain length were determined by the enzyme specificity, resulting in a difference in antibacterial efficiency. It agreed with Verma et al. [26], who discovered that porcine liver hydrolysate produced by papain had comparable antibacterial activity against *S. aureus* and *E. coli* with inhibitory zones of 15–16 mm.

Larger amphiphilic molecules with a high cysteine content and a net positive charge in their active form are typical of effective antibacterial peptides [59]. According to Liu et al. [60], the presence of cysteine and/or aromatic residues (tryptophan, tyrosine, and phenylalanine) in peptides plays an important role in their antibacterial and antifungal properties. Peptide antibacterial activity is also influenced by amino acid sequence, chain length, molecular weight, charge, and secondary structure [61]. According to papain specificity, the presence of such cysteine and/or aromatic residues in generated peptides may have contributed to its excellent antibacterial activity. The PPH produced by papain may quickly interfere with the internal microbial cytoplasm membrane surface [16], resulting in the breakdown of the microbial cell membrane and, ultimately, the inability to survive [26,62]. There was no difference in the zone of inhibition in both microorganisms between fresh placenta and Flavourzyme-made PPH, indicating that the generated peptides catalyzed by Flavourzyme had a lower antibacterial effect (Table 3). Tejano et al [63] proposed that higher DH did not always correspond to higher bioactivity.

## Molecular weight distribution of PPH

Following hydrolysis, Alcalase, Flavourzyme, and papain catalyzed the degradation of polypeptides to smaller peptides. The PPH produced by Alcalase with the highest DH (36.55%) had the smallest molecular weight (Fig 5). This was consistent with the findings of Benjakul and Morrissey [64], as well as Liaset et al. [65], who discovered small molecular weight peptides at high DH produced by Alcalase. According to Bhaskar et al. [66], protein hydrolysate with desirable nutritional values should contain a high proportion of low-molecular-weight peptides and a low proportion of free amino acids. Peptides with varying molecular weights could be produced by different enzymes. Placental peptides produced by trypsin, chymotrypsin, and pepsin had MWs of 106 and 500 Da, 20 kDa, and greater than 7 kDa, respectively [14]. Teng et al. [7] reported that goat placenta peptides with molecular weights ranging from 3 to 15 kDa had the maximum antioxidant activity [7]. As a result, the antioxidant activity was found to be related to peptide molecular mass [67,68].

## Fourier transform infrared spectrometer (FTIR)

The presence of spectral regions in the amide I and II bands relates to the peptide backbone, which corresponds to a different secondary structure. The occurrence of proteolysis has also

been demonstrated by the reduction in the size of polypeptides [69]. Chen et al. [28] and Mohammadian and Madadlou [70] reported that the main bonds C-O and N-H on the polypeptide backbone were easily sensitive to hydrogen bonds. Furthermore, the structure of amide I is more sensitive to structural changes than the structure of amide II, which contribute to the formation or disruption of hydrogen bonds [28]. When compared to fresh placenta, the intensities of amide I and II of porcine PPH catalyzed by papain, Flavourzyme, and Alcalase increased. The findings suggested that enzymatic hydrolysate was involved in the formation of complexes between hydrogen bonds and peptide bonds during the hydrolysis of porcine placenta protein. Furthermore, the bands around 2800–3500 cm$^{-1}$ were clearly found in fresh porcine placenta, and their intensities were reduced after PPH production. It was reported that the spectrum from animal fat can be found at 2,925 and 2,854 cm$^{-1}$, which corresponded to the symmetric and asymmetric stretching vibrations of C-H bonds, respectively [71]. The findings were consistent with the presence of fat in the original porcine placenta (Table 1). These fats can be removed during the production of PPH, resulting in a decrease in PPH band intensities. Lozano et al. [71] reported that because the bands between 3,500 and 3,000 cm$^{-1}$ do not appear in the fat spectrum, they may be related to proteins. According to the findings, those bands appeared to be reduced during hydrolysis, which was associated with the higher peaks around 1,400–1,700 cm$^{-1}$, which represented newly formed peptides.

## Conclusion

Porcine placenta was high in protein and EAA, making it an ideal raw material for the production of enzymatic protein hydrolysate. The DH, free radical inhibition, reducing power, and metal chelation of PPH were all influenced by enzyme type, enzyme concentration, and hydrolysis time. Higher enzyme concentrations and longer hydrolysis times resulted in PPH with higher DH and antioxidative activities. Flavourzyme-derived PPH demonstrated excellent ABTS$^{•+}$ scavenging activity, whereas papain-derived PPH scavenged more effectively against the hydrophobic DPPH$^{•}$. Furthermore, Alcalase produced PPH with superior reducing power and metal chelating ability when compared to other enzymatic PPH. After pepsin treatment, the DPPH$^{•}$ scavenging activity and reducing power of all PPH increased, then decreased after pancretin digestion, but the ABTS$^{•+}$ scavenging activity showed the opposite trend. The PPH produced by papain had higher antibacterial activity against *S. aureus* and *E. coli*. As a result, PPH can be thought of as an antioxidative and antibacterial agent that could be used in functional food products.

## Supporting information

**S1 Fig. Effects of enzyme type (Alcalase (a), Flavourzyme (b), and papain (c)), enzyme concentration, and hydrolysis time on the degree of hydrolysis (DH) of porcine placenta protein.**
(PPTX)

**S2 Fig. Effects of enzyme type (Alcalase (a, d), Flavourzyme (b, e), and papain (c, f)), enzyme concentration, and hydrolysis time on the DPPH$^{•}$ scavenging activity (a, b, c) and ABTS$^{•+}$ scavenging activity (d, e, f) of porcine placenta hydrolysate (PPH).**
(PPTX)

**S3 Fig. Effects of enzyme type (Alcalase (a, d), Flavourzyme (b, e), and papain (c, f)), enzyme concentration, and hydrolysis time on the reducing power (a, b, c) and metal chelating activity (d, e, f) of porcine placenta hydrolysate (PPH).**
(PPTX)

**S4 Fig. Changes in DPPH• scavenging activity (a), ABTS•+ scavenging activity (b), and reducing power (c) of porcine placenta hydrolysate (PPH), produced by Alcalase, Flavourzyme, and papain at 10% enzyme load for 20 min, in gastrointestinal (GI) tract model system.**
(PPTX)

**S5 Fig. Molecular weight distribution of fresh porcine placenta (brown line) and porcine placenta hydrolysate (PPH) produced by Alcalase (red line), Flavourzyme (blue line), and papain (green line) at 10% enzyme load for 20 min.**
(PPTX)

**S6 Fig. Fourier transform infrared (FTIR) spectra of freeze-dried porcine placenta (Fresh) and freeze-dried porcine placenta hydrolysate (PPH) produced by Alcalase, Flavourzyme, and papain at 10% enzyme load for 20 min.**
(PPTX)

**S1 Table. Proximate composition, TCA-soluble peptide, mineral profile, and amino acid composition of freeze-dried porcine placenta.**
(DOCX)

**S2 Table. Correlation coefficient of enzymatic hydrolytic variables and antioxidant activities of porcine placenta hydrolysate.**
(DOCX)

**S3 Table. Antibacterial activity (zone of inhibition) of porcine placenta hydrolysate (PPH) catalyzed by three different enzymes at 10% enzyme concentration for 20 min.**
(DOCX)

**S1 Dataset.**
(ZIP)

## Acknowledgments

We would like to thank Shaw Kaset Rungrueng Co. Ltd., Nakhon Si Thammarat, Thailand for supplying porcine placenta throughout the study.

## Author Contributions

**Conceptualization:** Phanthipha Laosam, Worawan Panpipat, Gorawit Yusakul, Ling-Zhi Cheong, Manat Chaijan.

**Data curation:** Phanthipha Laosam.

**Funding acquisition:** Worawan Panpipat.

**Investigation:** Phanthipha Laosam.

**Methodology:** Worawan Panpipat, Gorawit Yusakul, Ling-Zhi Cheong, Manat Chaijan.

**Supervision:** Worawan Panpipat.

**Writing – original draft:** Phanthipha Laosam, Worawan Panpipat, Manat Chaijan.

**Writing – review & editing:** Phanthipha Laosam, Worawan Panpipat, Gorawit Yusakul, Ling-Zhi Cheong, Manat Chaijan.

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
