## [Decision Letter · Decision Letter 0]

24 Aug 2021

PONE-D-21-18837

Porcine placenta hydrolysate as an alternate functional food ingredient

PLOS ONE

Dear Dr. Panpipat,

Thank you for submitting your manuscript to PLOS ONE. After careful consideration, we feel that it has merit but does not fully meet PLOS ONE’s publication criteria as it currently stands. Therefore, we invite you to submit a revised version of the manuscript that addresses the points raised during the review process.

We look forward to receiving your revised manuscript.

Kind regards,

Saikat Dewanjee

Academic Editor

PLOS ONE

Journal Requirements:

Additional Editor Comments (if provided):

After careful reading of this manuscript and considering the reviewers' response, I found the manuscript has merit for considering; however, the data and experiments are not adequate to substantiate the claim. Reviewers were also critical of the technical quality of this manuscript. Thus, it needs major revision. Additional experiments are required to substantiate the claim. I also suggest including more references from the same journal and replace old references.

Reviewers' comments:

Reviewer's Responses to Questions

**Comments to the Author**

1. Is the manuscript technically sound, and do the data support the conclusions?

Reviewer #1: Yes

Reviewer #2: Yes

Reviewer #3: Partly

Reviewer #4: Partly

2. Has the statistical analysis been performed appropriately and rigorously? 

Reviewer #1: Yes

Reviewer #2: Yes

Reviewer #3: N/A

Reviewer #4: Yes

3. Have the authors made all data underlying the findings in their manuscript fully available?

Reviewer #1: Yes

Reviewer #2: Yes

Reviewer #3: Yes

Reviewer #4: No

4. Is the manuscript presented in an intelligible fashion and written in standard English?

Reviewer #1: Yes

Reviewer #2: Yes

Reviewer #3: No

Reviewer #4: No

5. Review Comments to the Author

Reviewer #1: The presented manuscript, "Porcine placenta hydrolysate as an alternate functional food ingredient" seems okay. It will be very good if the authors go for thorough language checking prior publishing the article.

Reviewer #2: The manuscript entitled "Porcine placenta hydrolysate as an alternate functional food ingredient" describes the food functional values and antioxidant activities of Porcine placenta hydrolysate. The manuscript has been nicely represented with significant scientific work. Some figures are not clear to understand properly. English editing is needed since there are many Grammatical errors in the manuscript. Statistical significance should be included in the figure 1 to figure 4. After these changes, manuscript may be accepted for the publication.

Reviewer #3: The work done by Worawan Panpipat et al., entitled “Porcine placenta hydrolysate as an alternate functional food ingredient” is too preliminary for any scientific conclusion. There are few comments that need to be properly addressed before further progress of the manuscript.

1.The author should find out the effect of hydrolysate in invivo/ invitro model.

2.The author should also perform some additional experiment to established anti-microbial and antioxidative properties.

3.The author should provide appropriate figure legends.

4.Author should go through the manuscript thoroughly to check the typographical errors and

grammatical mistakes.

Reviewer #4: 1. Restructure Ln 52, 53, 54

2. Ln 56, 57 do not have citation

3. Ln 63, 63 has poor English, need correction

4. rationale of the study is not appropriate in the given interest

5. Mention Cat# for all the reagents/Chemicals, and other consumables.

6. All experiments were carried out in triplicate (N = 3), does it satisfy power of the study

7. Figure Legends needs to re-write and not conveying required information

8. anti-microbial and anti-oxidant properties are not established properly

6. PLOS authors have the option to publish the peer review history of their article (what does this mean?). If published, this will include your full peer review and any attached files.

Reviewer #1: No

Reviewer #2: No

Reviewer #3: No

Reviewer #4: No

---

## [Author Response · Author response to Decision Letter 0]

28 Aug 2021

Response to Reviewers

All points raised by the reviewers were carefully addressed and answered point-by-point. A revision was made in highlighted red fonts. The revised manuscript was carefully prepared to meet PLOS ONE's style requirements.

Journal Requirements:

Ans: The revised manuscript was carefully prepared to meet PLOS ONE's style requirements.

Ans: The ORCID iD for the corresponding author is given.

Ans: QuillBot, a paraphrase tool, was used to double-check the language throughout the manuscript.

Reviewer #1: The presented manuscript, "Porcine placenta hydrolysate as an alternate functional food ingredient" seems okay. It will be very good if the authors go for thorough language checking prior publishing the article.

Ans: QuillBot, a paraphrase tool, was used to double-check the language.

Reviewer #2: The manuscript entitled "Porcine placenta hydrolysate as an alternate functional food ingredient" describes the food functional values and antioxidant activities of Porcine placenta hydrolysate. The manuscript has been nicely represented with significant scientific work. Some figures are not clear to understand properly. English editing is needed since there are many Grammatical errors in the manuscript. Statistical significance should be included in the figure 1 to figure 4. After these changes, manuscript may be accepted for the publication.

Ans: The Preflight Analysis and Conversion Engine (PACE) digital diagnostic tool, https://pacev2.apexcovantage.com/, was used to check the quality of all figures. We double-checked that the figures complied with PLOS guidelines. QuillBot, a paraphrase tool, was used to double-check the language. All figures and tables were statistically analyzed, as stated in the text. The letters indicating statistical differences, on the other hand, were not included in the figures since they would make the figures too cluttered.

Reviewer #3: The work done by Worawan Panpipat et al., entitled “Porcine placenta hydrolysate as an alternate functional food ingredient” is too preliminary for any scientific conclusion. There are few comments that need to be properly addressed before further progress of the manuscript.

1.The author should find out the effect of hydrolysate in in vivo/ in vitro model.

Ans: The statement and its significance were given in the Abstract. “The production of bioactive peptides from animal-based raw materials highly depends on enzymatic hydrolysis. Porcine placenta is an underutilized biomass in Thailand's pig farms, yet it is still a source of proteins and beneficial compounds. Porcine placenta could be used as a protein substrate for the production of enzymatic hydrolysate, which could be employed as a functional food ingredient in the future. The goal of this study was to enzymatically produce porcine placenta hydrolysates (PPH) using three commercial enzymes (Alcalase, Flavouzyme, and papain) and evaluate their in vitro antioxidant and antibacterial activity.”. As a result, the antioxidant and antibacterial properties were investigated in vitro in this work. This study did not contain any in vivo testing. So the title was changed to “Porcine placenta hydrolysate as an alternate functional food ingredient: in vitro antioxidant and antibacterial assessments” to make it more apparent. In vitro tests can be performed to examine the bioactivity of active compounds such as protein hydrolysate, which is particularly useful in the area of Food Science and Technology.

2. The author should also perform some additional experiment to established anti-microbial and antioxidative properties.

Ans: The antibactial and antioxidant capabilities were tested and reported in the original manuscript. The antioxidant activities were exhibited, which included DPPH•/ABTS•+ scavenging activities (Fig. 2), reducing power (Fig. 3) and metal chelating activity (Fig. 3). Table 2 also includes the correlation coefficient between enzymatic hydrolytic variables and antioxidant activities of porcine placenta hydrolysate. In addition, the gastrointestinal (GI) tract model system was used to investigate changes in DPPH•/ABTS•+ scavenging activities and reducing power. The antibacterial activity of PPH was examined using the agar-well diffusion method against Staphylococcus aureus and Escherichia coli, and the zone of inhibition was reported. All of the tests can be used to draw conclusions about porcine placenta hydrolysate's antioxidant and antibacterial activities.

3.The author should provide appropriate figure legends.

Ans. All of the legends for the figures were double-checked.

4.Author should go through the manuscript thoroughly to check the typographical errors and

grammatical mistakes.

Ans: QuillBot, a paraphrase tool, was used to check for typographical errors and recheck the language.

Reviewer #4: 

1. Restructure Ln 52, 53, 54

Ans: It was changed to “Wastes and by-products are being reassembled in the farrow-to-finish production pig farming system, which consists of breeding pigs, producing piglets, and fattening pigs, as a result of the expansion of large-scale pig farms [3].”

2. Ln 56, 57 do not have citation

Ans: Because no research has been documented in Thailand, no references were provided. Therefore, the sentence was changed to “However, there has been no published research on the use of pig placenta in Thailand. It is of great interest to figure out how to use the right technologies to add value to this placenta.”

3. Ln 63, 63 has poor English, need correction

Ans: It was changed to “Collagen is the most abundant protein in the extracellular matrix of the placenta, with lately high market values [4, 5, 6].”

4. rationale of the study is not appropriate in the given interest

Ans: The rationale of the study was intensively revised as highlighted in the Introduction.

5. Mention Cat# for all the reagents/Chemicals, and other consumables.

Ans: It was stated in the “Chemical” section that “All enzymes (Alcalase from Bacillus licheniformis, Flavourzyme from Aspergillus oryzae protease, papain from papaya latex) and chemicals (e.g., trichloroacetic acid (TCA), 2,4,6-trinitrobenzene sulfonic acid (TNBS), 2,2-diphenyl-1-picrylhydrazyl (DPPH), 3-(2-pyridyl)-5-6-diphenyl-1,2,4-triazine-4′,4′′-disulphonic acid sodium salt (ferrozine), 2,2-azino-bis(3-thylbenzothiazoline-6-sulphonicacid) diammonium salt (ABTS), 2,4,6-tripyridyl-s-triazine (TPTZ)) were purchased from Sigma-Aldrich (St. Louis, MO, USA).”. It met the journal's requirements, and many standard journals do not require the Cat#. As a result, we didn't include the Cat#. Thank you very much.

6. All experiments were carried out in triplicate (N = 3), does it satisfy power of the study

Ans: All experiments were run in triplicate with three different lots of placenta (N = 3). As a result, the study's power was satisfied.

7. Figure Legends needs to re-write and not conveying required information

Ans: The figure legends were revised accordingly.

8. anti-microbial and anti-oxidant properties are not established properly

Ans: The antioxidant activities were tested, which included DPPH•/ABTS•+ scavenging activities (Fig. 2), reducing power (Fig. 3) and metal chelating activity (Fig. 3). Table 2 also includes the correlation coefficient between enzymatic hydrolytic variables and antioxidant activities of porcine placenta hydrolysate. In addition, the gastrointestinal (GI) tract model system was used to investigate changes in DPPH•/ABTS•+ scavenging activities and reducing power. The antibacterial activity of PPH was examined using the agar-well diffusion method against Staphylococcus aureus and Escherichia coli, and the zone of inhibition was reported. All of the tests can be used to draw conclusions about porcine placenta hydrolysate's antioxidant and antibacterial activities.

Ans: Done.

---

## [Decision Letter · Decision Letter 1]

28 Sep 2021

Porcine placenta hydrolysate as an alternate functional food ingredient: in vitro antioxidant and antibacterial assessments

PONE-D-21-18837R1

Dear Dr. Panpipat,

We’re pleased to inform you that your manuscript has been judged scientifically suitable for publication and will be formally accepted for publication once it meets all outstanding technical requirements.

Kind regards,

Saikat Dewanjee

Academic Editor

PLOS ONE

Additional Editor Comments (optional):

Reviewers' comments:

Reviewer's Responses to Questions

**Comments to the Author**

1. If the authors have adequately addressed your comments raised in a previous round of review and you feel that this manuscript is now acceptable for publication, you may indicate that here to bypass the “Comments to the Author” section, enter your conflict of interest statement in the “Confidential to Editor” section, and submit your "Accept" recommendation.

Reviewer #2: All comments have been addressed

Reviewer #3: All comments have been addressed

2. Is the manuscript technically sound, and do the data support the conclusions?

Reviewer #2: Yes

Reviewer #3: Yes

3. Has the statistical analysis been performed appropriately and rigorously? 

Reviewer #2: Yes

Reviewer #3: I Don't Know

4. Have the authors made all data underlying the findings in their manuscript fully available?

Reviewer #2: (No Response)

Reviewer #3: (No Response)

5. Is the manuscript presented in an intelligible fashion and written in standard English?

Reviewer #2: Yes

Reviewer #3: No

6. Review Comments to the Author

Reviewer #2: (No Response)

Reviewer #3: (No Response)

7. PLOS authors have the option to publish the peer review history of their article (what does this mean?). If published, this will include your full peer review and any attached files.

Reviewer #2: No

Reviewer #3: **Yes: **Parames C. Sil

---

## [Editor Report · Acceptance letter]

7 Oct 2021

PONE-D-21-18837R1 

Porcine placenta hydrolysate as an alternate functional food ingredient: *in vitro* antioxidant and antibacterial assessments 

Dear Dr. Panpipat:

I'm pleased to inform you that your manuscript has been deemed suitable for publication in PLOS ONE. Congratulations! Your manuscript is now with our production department. 

Kind regards, 

on behalf of

Dr Saikat Dewanjee 

Academic Editor

PLOS ONE